# Effect of Phytase Derived from the *E. coli* AppA Gene on Weaned Piglet Performance, Apparent Total Tract Digestibility and Bone Mineralization

**DOI:** 10.3390/ani10010121

**Published:** 2020-01-11

**Authors:** Zuzanna Wiśniewska, Lode Nollet, Anouk Lanckriet, Erik Vanderbeke, Spas Petkov, Nikolay Outchkourov, Małgorzata Kasprowicz-Potocka, Anita Zaworska-Zakrzewska, Sebastian A. Kaczmarek

**Affiliations:** 1Poznan University of Life Sciences, Department of Animal Nutrition, 60-637 Poznań, Wołyńska 33, Poland; zuzanna.wisniewska@up.poznan.pl (Z.W.); malgorzata.potocka@up.poznan.pl (M.K.-P.); anita.zaworska-zakrzewska@up.poznan.pl (A.Z.-Z.); 2Huvepharma, Uitbreidingstraat 80, 2600 Antwerpen, Belgium; Lode.Nollet@huvepharma.com (L.N.); Anouk.Lanckriet@huvepharma.com (A.L.); Erik.Vanderbeke@huvepharma.com (E.V.); 3Huvepharma EOOD, Nikolay Haytov Str., 3a, 1113 Sofia, Bulgaria; Spas.Petkov@huvepharma.com (S.P.); nikolay.outchkourov@huvepharma.com (N.O.)

**Keywords:** phytase, P apparent total tract digestibility, P digestibility, Ca digestibility, bone mineralization, weaned piglets performance

## Abstract

**Simple Summary:**

Plants store phosphorus mainly in a form that is indigestible to pigs: phytate. Unabsorbed minerals excreted into the environment are considered to be environmental pollutants. Our study focuses on the effects of novel types of phytase enzymes in diets deficient in external phosphorus on the performance, digestibility of crude protein, dry matter and minerals, and bone mineralization of weaned piglets. Using novel phytases, we noted a positive impact on body weight gain, feed conversion ratio, the digestibility of phosphorus and calcium, and phosphorus and calcium content in bones. There was no significant improvement in the content of dry matter or digestion of crude protein. This study reveals novel phytases to be useful additives for weaned piglet diets because of their superior performance results and phosphorus absorption.

**Abstract:**

The objective of this study was to assess the effect of novel appAT1 and appAT2 phytase inclusion at 250 phytase units (FTU)/kg on weaned piglet performance, the apparent total tract digestibility of P and Ca, and bone mineralization. Piglets (48 males) were randomly divided into four treatment groups: a positive control (PC), with recommended levels of phosphorus (P) and calcium (Ca), a negative control (NC) deficient in P and Ca, and two experimental groups with NC diets supplemented with phytase derived from the appA gene of *Escherichia coli*. Diets fed in a mashed form were divided into prestarter (0–21 days) and starter (22–42 days) periods. During the whole period of the study, the experimental diets improved (*p* < 0.05) the body weight gain (BWG) and feed conversion ratio (FCR) compared to the NC diet. The apparent total tract digestibility (ATTD) of the dry matter and crude protein was not significantly different among the diets. Phytase-supplemented diets improved the ATTD of P (*p* < 0.05) and the ATTD of Ca (*p* < 0.05). Bone ash content in the third metacarpal and P and Ca content were improved among the phytase supplemented diets compared to the NC diet.

## 1. Introduction

Phosphorus (P) is an irreplaceable macroelement that regulates the metabolic and physiological functions of a living organism. It is known to take part in skeletal mineralization, as well as the formation of cell membranes, nucleic acids, and adenosine triphosphate [1]. Plants store P mainly in the form of phytate, which is not completely digestible by monogastric animals. Therefore, unabsorbed P is excreted into the environment where it is considered to be pollution and an environmental hazard. Commercial pigs fed diets with no phytase enzymes excrete over 50–80% of their total P intake [2]. Moreover, as an essential nutrient, especially for growing animals, P is one of the most expensive feed components that provide an inorganic form of phosphates (30–50% of the total P), which are only partially digested (approximately 68%) [3]. Thus, avoiding P losses seems to be reasonable from nutritional, economic, and environmental perspectives.

The activities of endogenous and intestinal microbial phytases (the enzymes responsible for the dephosphorylation of phytate) are limited and cover only a negligible part of the P requirements for animals [4]. In pigs, mucosal phytase hydrolyzes only the lower esters of the phytate molecule [5], mostly in the jejunum. Microbial phytase of the gut can degrade phytate; however, Sandberg et al. [6] reported that, in the large intestine, phytate degradation is affected by the presence of calcium (Ca). Phytase is also present in several plants, such as wheat, barley, rye, triticale, and soybean seeds [4]. Nevertheless, under the high temperatures maintained for the technological treatments of feed, the activity of plant phytase is inhibited [7].

Over the years, the application of different exogenous phytases at various levels (FTU/kg) has shown favorable effects on nutrient digestibility. The first commercially available phytase was 3-phytase derived from *Aspergillus niger*, but in the last two decades, 6-phytase originating from *Escherichia coli*, *Peniophora*, *Citrobacter*, or *Buttiauxella* has also been available on the market. Adeola et al. [8], in a study on young pigs and the usage of phytase derived from *Escherichia coli* and *Peniophora lycii* at 500 or 1000 FTU/kg, reported increased daily gains after supplementing the enzymes in the negative control diet with low inorganic P content. Moreover, the inclusion of 500 FTU/kg of phytase derived from *Peniophora lycii* released 0.572 g/kg of P, whereas *E. coli* phytase released 0.770 g/kg of P and enhanced metacarpal bones mineralization. There were no significant differences among the digestibility of dry matter, protein, and energy. Similarly, increased ATTD of P (27.9%) and Ca (22.7%) were reported in older pigs fed diets containing various protein sources (soybean meal or sesame meal) with additional phytase at a 1000 FTU/kg feed level [9].

In the present study, two novel variants of phytases with highly improved thermal stability were obtained via protein engineering from the appA gene of *E. coli* [10]. Both strains differ in speed of phytate degradation and affinity for phytate (V_max_ and K_M_, respectively; in vitro Michaelis–Menten Kinetics). The objective of this study was to assess the effect of inclusion of appAT1 and appAT2 phytases at 250 FTU/kg in diets on weaned piglet performance, the apparent total tract digestibility (ATTD) of P and Ca, and bone mineralization.

## 2. Materials and Methods

### 2.1. Animals and Diets

All the experimental procedures complied with the guidelines of the Local Ethical Committee for Experiments on Animals in Poznan regarding animal experimentation and animal care under study (European Union (EU) Directive 2010/63/EU for animal experiments). Individual approval for this trial was not required because of the production standards used during this study, and all samples were collected post-slaughter or during the study but without a negative impact on animal welfare (excreta collection).

A total of 48 castrated male weaned piglets (Naima × (Pietrain × Duroc)), aged 28 days with body weights (BWs) of around 9.95 kg, were raised on floor pens. Prior to delivery of the piglets, all pens within the rooms were thoroughly pressure-washed and suitably disinfected, according to usual farm practice. The piglets received all necessary veterinary vaccinations and had unlimited access to water and feed. The study was designed as a complete randomized with 12 replications of four diets: a positive control (PC), negative control (NC), and two experimental NC diets with the addition of phytases: appAT1 phytase and appAT2 phytase, dosed at 250 FTU/kg. These two thermostable phytases were produced by Huvepharma based on the appA2 gene of *E. coli*, as described in the study of Rodriguez et al. [10]. The phytases were mixed with wheat flour and a starch carrier to facilitate their inclusion in the experimental diets. The study lasted 42 days and was divided into two periods: the prestarter period (1–21 days) and the starter period (22–42 days). The composition of the diets is presented in Table 1 and Table 2. The diets were formulated to meet or exceed the NRC requirements [11], except for Ca and P in NC diets with or without phytase addition. The starter diet contained titanium dioxide (TiO_2_) as an indigestible marker. Differences in digestible P and Ca were targeted to be 0.1% of the total Ca for the NC diet and 0.16% of the total Ca for the PC diet, which was realized by the formulation of diets with different limestone and monocalcium phosphate proportions. The prestarter PC diet contained 0.75% Ca, 0.61% total P, and 0.32% digestible P, while the NC diet contained 0.59% Ca, 0.51% total P, and 0.22% digestible P (Table 1). The starter PC diet contained 0.73% Ca, 0.60% total P, and 0.30% digestible P, while the starter NC diet contained 0.57% Ca, 0.49% total P, and 0.20% digestible P. The basal diet was prepared in a horizontal mixer (Zuptor 100, 100 kg capacity) under 4 min of mixing, with the mixing band set at 27.4 rev/min. The compounds were ground with discs set apart at a distance of 1.8 mm (Skiold Disk Mill SK2500, Skiold A/S, Sæby, Denmark). Minerals with amino acids, vitamins, and fat were directly added to the mixer. After the preparation of each diet, the mixer was cleaned with wheat flour for 10 min to avoid contamination.

Each diet was randomly allocated to 12 pens, with one piglet per pen. Every pen had a separate feed crate. All pens had free access to feed hoppers and water drinkers with the mains water provided ad libitum via a header tank. All diets were offered in a mashed form, ad libitum. The health status and condition of each piglet were observed twice a day by qualified personnel. The BW of each piglet, as well as the feed intake (FI), was recorded at 21 and 42 days to calculate the feed conversion ratio (FCR).

### 2.2. Digestibility

On days 34–38, feces were collected twice daily (around 8 AM and 4 PM) from Monday to Friday and immediately frozen (−20 °C). At the end of the study, the materials were pooled for each animal, and subsamples were taken for analysis. During the days of feces sampling, feed samples were taken from each diet and blended into one pooled sample.

### 2.3. Bone Ash

At the end of the study, 12 piglets from each group were stunned by electric shock and killed by exsanguination to obtain the third metacarpals from the right foot. The metacarpals were boiled to remove their tissues and cartilage caps. Ground dried bone samples were extracted to remove fat. Then, the extracted samples were burned in a muffle furnace (P330, Nabertherm GmbH, Lilienthal, Germany) at 600 °C for 5 h, and the ash weight of each sample was expressed as a percentage of the dry fat-extracted bone weight.

### 2.4. Analyses

Three feed samples of 500 g were collected per diet during manufacture and blended to obtain a pooled sample for analysis. The diets from both periods (prestarter and starter) were analyzed in duplicate to estimate the content of dry matter, crude protein, crude fiber, crude fat, crude ash, P, and Ca, according to the procedure of the Association of Official Agricultural Chemists (AOAC) [12]. A Kjel Foss Automatic 16,210 (A/S N. Foss Electric, Denmark) was used to examine the nitrogen content. Fat content was determined using a Soxtec System HT 1043 Extraction Unit (Foss Tecator, Denmark). The pooled feed samples obtained for the evaluation of digestibility, and fecal samples, were analyzed to estimate the content of crude protein, dry matter, Ca, and P, according to the procedure of the AOAC [12]. The TiO_2_ content was determined by the method of Short et al. [13], as modified by Myers et al. [14].

Phytase activity was spectrophotometrically analyzed using a Biovet (Bulgaria), according to an EN ISO 30,024 phytase assay. The proportion of the phytate-P in diets was determined using the method described by Reichwald and Hatzack [15].

The P ATTD was calculated according to the following formula:ATTD (%) = {1 − [(IM% diet/IM% excreta) × (P% excreta/P% diet)]}(1)
where IM is the content of dietary marker (TiO_2_).

Forty-eight third metacarpals were boiled to remove their tissues and cartilage caps and weighed separately. Ground (in a 0.1 mm sieve) and dried bone samples were extracted with ether to remove fat, after which they were ashed in a muffle furnace (P330, Nabertherm GmbH, Lilienthal, Germany) at 600 °C for 5 h. The ash weight of each sample was expressed as a percentage of the dry fat-extracted bone weight [16]. The content of P and Ca in the bone ash was determined according to the procedure of AOAC [17].

### 2.5. Statistical Analysis

The growth performance, digestibility, and bone parameters were estimated using an analysis of variance with GLM procedure in the “Agricolae” package [18] of the R program [19]. The parameters were determined according to the following formula:*Y_ij_* = µ + *α_i_* + *e_ij_*(2)
where *Y_ij_* is the observed dependent variable, *α_i_* is the effect of the diet, and *e_ij_* is the random error within the model, with significance set at *p* < 0.05. Tukey’s multiple comparisons were used to determine differences among the means.

## 3. Results

### 3.1. Diets

The Weende analysis of the prestarter and starter diets is shown in Table 2. It can be observed that the analyzed values are close to the formulated values, except that of the total P in the starter PC diet, which was 0.06 percent point above the formulated value. The target dosage of phytase was confirmed by analysis (Table 2).

Analysis of the feed samples taken from the starter diet to evaluate digestibility showed no difference between the feed samples and the formulated feed, except for the total P in the PC diet, which was 0.06 percent point (Table 3).

### 3.2. Animal Performance

There were no differences in the health status of piglets receiving different diets, and no mortality was noted during the study. The consistency of the feces was normal during the whole study, and no complications, such as sticky droppings or diarrhea, were observed.

Piglets fed the NC diet had the lowest final BW (36.9 kg) compared to the piglets receiving other diets numerically. However, this weight was not significantly different from that of the PC (38.5 kg) (Table 4). The NC diet supplemented with phytase appAT1 or appAT2 resulted in a significantly greater final BW than the NC diet (39.4, 40.0, and 36.9 kg for NC + appAT1, NC + appAT2, and NC, respectively; *p* < 0.05). The body weight gain (BWG) in the prestarter period (days 0–21) was significantly lower (*p* < 0.05) in the NC diet compared to other diets, but not during the starter period (days 21–42). Over the entire study, the BWG was the lowest in the NC diet, which was significantly different from the BWG of the piglets receiving other diets (*p* < 0.05).

The feed intake (FI) was not significantly different between the different diets during the periods, but numerically the greatest FI was observed in the NC diet (Table 4).

Animals receiving the NC diet had the greatest FCR, which was significant (*p* < 0.05) during the prestarter period and the entire study (2.202 vs. 2.010, 1.929, and 1.916 for NC, PC, NC + appAT1, and NC + appAT2, respectively) (Table 4). Pigs fed phytase-supplemented diets had a PC had similar FCR.

### 3.3. Digestibility

The ATTD of the dry matter and crude protein was not affected by diet type (Table 5). The P ATTD was the lowest in piglets fed a NC diet (*p* < 0.05 vs. other treatments; 27.6% vs. 40.7%, 46.8%, and 48.4% for NC, PC, NC + appAT1, and NC + appAT2, respectively; Table 5).

There were no significant differences in the P ATTD between piglets fed the PC, NC+appAT1, and NC + appAT2 diets, although the P ATTD of piglets fed experimental diets showed a numerically greater P ATTD than those fed a PC diet (6.1% for NC + appAT1 and 7.7% NC + appAT2). Similarly, there was a trend (*p* = 0.069)toward a greater CP ATTD among the experimental diets. The lowest Ca ATTD was determined for piglets fed the unsupplemented NC diet (62.9%), but there were no differences with the PC (64.4%) diet (*p* > 0.05). Both phytase-supplemented NC diets were characterized by the highest Ca ATTD, which was significantly different from that of the NC and PC diets (*p* < 0.05).

### 3.4. Bone Ash

Analyses of bone ash in the third metacarpals (Table 6) showed that piglets fed diets without phytase (NC) were characterized by the lowest amount of bone ash (*p* < 0.05 vs. other treatments). The NC diet supplemented with phytases (NC + appAT1 or NC + appAT2) improved the amount of bone ash compared to that of the NC diet piglets (*p* < 0.05) and brought the level equal to, or above, that of the PC diet. There were no significant differences between total ash weights across diets, although numerically the lowest ash weight was observed for the unsupplemented NC diet. The content of Ca in the ash was the highest for the NC diets supplemented with phytases, which was significantly greater than that of the PC and NC diets (*p* < 0.05). Improved P content (*p* < 0.1) was found in the metacarpal ash with the phytase appAT2-supplemented diet compared to the NC diet.

## 4. Discussion

The inclusion of phytase to diets of monogastric animals has been proven to increase their performance [20,21,22], albeit sometimes only at high levels of supplementation (>1000 FTU/kg; [23]). This finding is in agreement with the study of Madrid et al. [24], which showed that the final BW was significantly greater in animals receiving a control diet with the recommended P level or a diet with low P and 500 FTU/kg of phytase (Finase Pc, AB Enzymes GmbH, Darmstadt, Germany) compared to those receiving a diet with a reduced P level. However, in their study, no significant differences in FCR were found, although the diet with low P and 500 FTU/kg of phytase numerically had the greatest FCR. In the current study, significant differences were identified in the BWG and FCR between the NC diet and the NC diet supplemented with two novel phytases, appAT1 and appAT2, even at 250 FTU/kg, although the digestible P-value of the used NC diets was not extremely low. Differences in response to growth might be linked to the activity of the phytase used, as observed by Guggenbuhl et al. [25], who showed that a greater inclusion of *E. coli* phytase was needed to obtain a significant increase in the daily gain of weaned piglets compared to *Citrobacter braakii* phytase or *Buttauxiella* phytase (diet containing 0.27% phytic-P and 0.55% total P).

As phytate has the ability to form a complex with proteins and amino acids, phytase addition can enhance protein digestibility due to its dephosphorylation of phytate [1]. The current study detected a numerical improvement in the ATTD of dry matter and crude protein in an NC diet supplemented with appAT2 phytase. However, no statistically significant differences were observed compared to other diets. This is in agreement with a study on poultry, in which the addition of a phytase product (RONOZYME HiPhos) at 5000 U/g to the diets (at 0.04%) containing different protein sources (soybean meal, yellow lupin meal, and blue lupin meal) did not affect the ATTD of the protein [16]. Similarly, Cervantes et al. [26] conducted a study on swine fed with sorghum–soybean meal diets supplemented with phytase and observed no improvement in the ATTD of amino acids. Zeng et al. [27] found that the ATTD of crude protein was significantly increased in piglets (20–25 kg) fed a diet with 0.14% non-phytate-P and 0.38% total P only at a concentration of 20,000 FTU/kg and not at 500 or 1000 FTU/kg. In the study of Dersjant-Li et al. [23], supplementation of *Buttiauxella* phytase at 500, 1000, or 2000 FTU/kg to a P-deficient corn-soy diet (2 g/kg digestible P) showed improvements (*p* < 0.1) in nitrogen (and dry matter) ATTD in piglets aged 21–28 days.

Ambiguous results may have been obtained due to the complexity of factors that determine the formation of bonds between phytate-P and protein, as these bonds by the type and solubility of the protein or the pH value [1].

The affinity of Ca and multivalent cations to greater phosphorylated inositol phosphates (InsP6) ensures the formation of phytate–mineral insoluble bonds [1]. Hence, if diets contain a relatively high amount of phytate, the absorption of Ca, which is essential for good bone mineralization, especially during the first 12 weeks of a pig’s life [28], will be impaired. This might lead to impaired bone mineralization, resulting in lameness and other skeletal irregularities and contributing to the acceleration of the culling rate, which is not desirable from an economic and animal welfare point of view. Furthermore, enhancement of the excretion of P to the environment as a result of the inability of monogastric animals to dephosphorylate phytate is considered to be a pollution factor leading to water eutrophication. Over the years, phytase addition has been shown to mitigate the negative effects of phytate on the digestion of Ca, P, and minerals [24,29,30], as well as P ATTD [23]. The current study is in agreement with the previous assessments showing the ability of phytase to hydrolyze phytate, thereby improving the digestibility of Ca and P. This ability also increases linearly with increasing doses of phytase, as observed by Dersjant-Li et al. [23], who used levels up to 2000 FTU/kg of feed containing 0.14% non-phytate-P and 0.38% total P, and by Zeng et al. [27], who used levels up to 20,000 FTU/kg.

In this study, the NC diet deficient in P, without any supplementation of phytases, had the lowest amount of bone ash, including the content of Ca and P in the ash. Numerically, the most favorable content of Ca and P in the ash was achieved for diets supplemented with phytase. These results are compatible with those obtained by Varley et al. [30], who showed that both the amount of bone ash and the content of Ca and P in bone ash are enhanced with a higher level of phytase addition (1000 or 1500 FTU/kg). On the one hand, the study of Kühn et al. [31], in which 500 and 2000 FTU/kg of an *E. coli* phytase was added to the diets of growing piglets containing 3.6–3.8 g/kg total P, showed a significant increase in the amount of rib ash, even at 500 FTU/kg. On the other hand, Madrid et al. [24] indicated that phytase inclusion had no influence on metacarpal mineralization (P and Ca) using a diet containing 4.5 g of total P. However, this discrepancy could be linked to the usage of older animals instead of weanling pigs. As already mentioned, bone formation occurs mainly during the first 12 weeks of a pig’s life [28], and there was already a high level of P in the diets used. Thus, the control diet was less P deficient. Additionally, Wu et al. [32] suggested that dietary P requirement by growing pigs is substantially overestimated in the currently used nutrient requirements of swine.

## 5. Conclusion

The addition of two novel phytases, appAT1 and appAT2, to weaned piglets’ diets deficient in Ca and P at a dose of 250 FTU/kg/diet significantly improved the final BW, BWG, and FCR. The obtained data showed that the addition of phytases to a phosphorus-deficient diet restored performance parameters to those supplemented with the recommended phosphorus level. The ATTD of Ca and P was also significantly increased to values observed in positive control pigs. However, no significant improvement in the dry matter or crude protein digestion was noted.

## Figures and Tables

**Table 1 animals-10-00121-t001:** Composition of the positive control (PC) and negative control (NC) diets and their calculated nutritional values.

Ingredients (%)	Basal Diets			
	Prestarter Diet (Days 1–21)		Starter Diet (Days 22–42)	
	PC	NC	PC	NC
Corn	68.32	69.52	70.04	70.91
Soybean meal (extracted)	27.09	26.93	25.96	25.78
Limestone	0.99	0.78	0.99	0.78
Monocalcium phosphate	1.13	0.68	1.09	0.59
Premix ^1^	0.50	0.50	0.50	0.50
Salt	0.50	0.50	0.50	0.50
Lysine-HCl	0.53	0.54	0.32	0.32
L-threonine	0.24	0.25	0.13	0.13
DL-methionine	0.23	0.23	0.13	0.13
L-tryptophan	0.06	0.07	0.05	0.05
Rapeseed oil	0.42		-	-
TiO_2_	-	-	0.30	0.30
Calculated nutrient content (%)				
ME (MJ/kg) ^2^	13.60	13.60	13.69	13.69
Crude protein	18.60	18.64	18.00	18.00
Crude fat	3.63	3.25	3.27	3.31
Crude fiber	3.25	3.27	3.23	3.24
Lysine (dig)	1.18	1.18	0.98	0.98
Methionine + cystine (dig)	0.71	0.71	0.61	0.61
Threonine (dig)	0.77	0.77	0.64	0.64
Tryptophan (dig)	0.21	0.21	0.19	0.19
Ca	0.75	0.59	0.73	0.57
P_total_	0.61	0.51	0.60	0.49
P_dig_	0.32	0.22	0.30	0.20

^1^ Mineral and vitamin premixed content, provided per kg of feed: Ca, 1.3 g; choline chloride, 200 mg; Fe, 75 mg; Cu, 20 mg; Co, 0.3 mg; Mn, 30 mg; Zn, 75 mg; I, 0.6 mg; Se, 0.15 mg; antioxidants (butylated hydroxyanisole, butylated hydroxytoluene); vitamin A, 7500 IU; vitamin D_3_, 1500 IU; vitamin E, 52.5 mg; vitamin K_3_, 1.1 mg; vitamin B_1_, 1.1 mg; vitamin B_2_, 3.0 mg; vitamin B_6_, 2.25 mg; pantothenic acid, 7.5 mg; nicotinic acid, 15 mg; folic acid, 1.5 mg; vitamin B_12_, 18.5 μg, biotin, 75 μg. ^2^ ME—metabolizable energy.

**Table 2 animals-10-00121-t002:** Chemical analysis (in %) and phytase activity (in FTU/kg) of the positive control (PC), negative control (NC), or NC supplemented with phytase (appAT1 or appAT2) for the prestarter and starter feed.

Items	Crude Fiber	Crude Protein	Crude Ash	Crude Fat	Ca	P	Phytase Activity
**Prestarter**							
PC	3.25	18.6	5.78	3.62	0.76	0.62	26
NC	3.29	18.6	5.14	3.23	0.57	0.51	33
NC + appAT1	3.30	18.6	5.10	3.24	0.57	0.49	280
NC + appAT2	3.31	18.7	5.12	3.26	0.58	0.50	290
**Starter**							
PC	3.23	17.7	4.65	3.30	0.71	0.66	17
NC	3.24	17.5	4.95	3.31	0.58	0.48	21
NC + appAT1	3.26	18.0	4.91	3.33	0.57	0.48	290
NC + appAT2	3.24	17.9	4.93	3.31	0.57	0.47	295

**Table 3 animals-10-00121-t003:** Ca, P, crude protein, and marker content (%) in the positive control (PC), negative control (NC), and NC diets supplemented with phytase (appAT1 or appAT2).

Diet	Crude Protein	TiO_2_	Ca	P
PC	17.26	0.312	0.74	0.66
NC	17.53	0.289	0.58	0.48
NC + appAT1	17.59	0.294	0.57	0.46
NC + appAT2	17.89	0.302	0.57	0.46

**Table 4 animals-10-00121-t004:** Initial body weight (IBW), final body weight (FBW), body weight gain (BWG), feed intake (FI), and feed conversion ratio (FCR) of piglets fed the positive control (PC), negative control (NC), or NC diets supplemented with phytase (appAT1 or appAT2; n = 12).

Diet	PC	NC	NC + appAT1	NC + appAT2	SEM	*p*
IBW (kg)	9.90	10.0	9.93	10.0	0.096	0.804
FBW (kg)	38.5 ^ab^	36.9 ^b^	39.4 ^a^	40.0 ^a^	0.313	0.013
BWG (kg/period)						
0–21 days	11.5 ^a^	10.2 ^b^	12.4 ^a^	12.6 ^a^	0.213	0.002
21–42 days	17.2	16.8	17.1	17.4	0.203	0.613
0–42 days	28.6 ^a^	26.9 ^b^	29.4 ^a^	30.0 ^a^	0.303	0.013
FI (kg)						
0–21 days	20.9	21.8	21.0	21.7	0.201	0.339
21–42 days	36.4	36.9	35.8	35.7	0.261	0.174
0–42 days	57.3	58.7	56.8	57.4	0.318	0.612
FCR (kg/kg)						
0–21 days	1.844 ^b^	2.164 ^a^	1.700 ^b^	1.733 ^b^	0.036	0.012
21–42 days	2.132	2.238	2.097	2.059	0.032	0.212
0–42 days	2.010 ^b^	2.202 ^a^	1.929 ^b^	1.916 ^b^	0.028	0.024

^ab^ Means in the columns with different letters are significantly different at *p* ≤ 0.05. SEM—standard error of the mean.

**Table 5 animals-10-00121-t005:** Apparent total tract digestibility (ATTD in %) of the dry matter (DM), crude protein (CP), P, and Ca of piglets fed the positive control (PC), negative control (NC), and NC diets supplemented with phytase (appAT1 or appAT2), (n = 12).

Diet	ATTD (%)			
	DM	CP	P	Ca
PC	77.0	63.0	40.7 ^a^	64.4 ^b^
NC	76.2	61.6	27.6 ^b^	62.9 ^b^
NC + appAT1	76.9	64.7	46.8 ^a^	79.1 ^a^
NC + appAT2	79.1	67.2	48.4 ^a^	82.4 ^a^
SEM	0.55	0.97	1.53	1.87
*p*	0.140	0.069	0.002	<0.001

^ab^ Means in the columns with different letters are significantly different at *p* ≤ 0.05. SEM—standard error of the mean.

**Table 6 animals-10-00121-t006:** Metacarpal bone characteristics of piglets fed the positive control (PC), negative control (NC), and NC diets supplemented with phytase (appAT1 or appAT2; n = 12).

Diet	Bone Ash (%)	Ash (g)	Ca (% of Ash)	P (% of Ash)
PC	33.8 ^a^	2.23	33.2 ^b^	16.3 ^AB^
NC	29.9 ^b^	1.76	32.8 ^b^	16.1 ^B^
NC + appAT1	34.3 ^a^	2.09	34.3 ^a^	16.3 ^AB^
NC + appAT2	34.6 ^a^	2.19	34.7 ^a^	16.5 ^A^
SEM	0.38	0.04	0.14	0.06
*p*	0.010	0.480	<0.001	0.060

^ab^ Means in the columns with different letters are significantly different at *p* ≤ 0.05. ^AB^ Means in the rows with different letters are significantly different at *p* ≤ 0.1. SEM—standard error of the mean.

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
