# Peer review of "Effect of Phytase Derived from the E. coli AppA Gene on Weaned Piglet Performance, Apparent Total Tract Digestibility and Bone Mineralization"

_animals, 2020, doi:10.3390/ani10010121_

Round 1
Reviewer 1 Report
This manuscript investigated the effect of two phytases in the Appa gene of E. Coli on weaned piglets. The properties of P, Ca, apparent total digestibility and bone mineralization were measured. The research reported is worthy of investigating and may be very interesting in this area. The manuscript reads well. Each part is well introduced and discussed. I recommend publishing it with minor revisions.
To improve the quality of the author's manuscript, some specific comments are as follows:
What is the types of effects of AppAT1 and AppAT2 phytase on phytate used in the experiment? What are the enzyme properties of AppAT1 and AppAT2? What is the difference between AppAT1 and AppAT2? This needs to be introducted. Line 95-97, the description of the content of digestible P is 0.22%, which is not match with the data in Table 1. Line 106, what is the ‘J’ represent? Line 242, it is inappropriate to pointed out that differences in response to growth might be linked to the source of phytase used. Many factors affect the growth of the piglets, and different protein source affect the activity of phytase. I suggest rewrite this sentence. Line 262-265, how did you come to this conclusion? Is there any reference? There are many formatting errors in this article, such as the writing of "P-value"(P or P), the writing of vitamin K3(K3 or K3) on line 108 and so on. Please check them carefully. Dietary components in each group are not equal to 100% in Table 1. Please correct the data. The titles and notes of Tables 3, 4 and 7 are not consistent with other table formats. Please check these issues such as "centering, line height, and first line indentation". There are no Tables 5 and 6 in this article. Some data do not meet the formula FCR = FI / BWG in Table 4. Isn't day 0-42 of the NC + AppAT1 column in Table 4 29.47 ≈ 29.5? Why is 29.47≈29.4? Please correct the data in the table. It is not recommended to place Table 7 on two pages, please typeset reasonably. Reference to ‘X. Wu, Z. Ruan, Y. G. Zhang, Y. Q. Hou, Y. L. Yin*, T. J. Li, R. L. Huang, W. Y. Chu, X. F. Kong, B. Gao, L. X. Chen. True digestibility of P in different resources of P ingredients. Asian-Australasian Journal of Animal Science, 2008, 21(1):107-119. ’
Author Response
Dear Reviewer,
The authors would like to thank the reviewer for a helpful critique of the manuscript. The manuscript has been improved to reflect the required changes and these modifications are detailed below and directly in the text using track-changes mode.
What are the types of effects of AppAT1 and AppAT2 phytase on phytate used in the experiment? What are the enzyme properties of AppAT1 and AppAT2? What is the difference between AppAT1 and AppAT2? This needs to be introduced.
Response: We have improved this and now we have some additional sentences:
In the present study, two novel variants of phytases with highly improved thermal stability were obtained via protein engineering from the appA gene of E. coli [10]. Both strains differ in speed of phytate degradation and affinity for phytate (Vmax and KM respectively; in vitro Michaelis-Menten Kinetics).
Line 95-97, the description of the content of digestible P is 0.22%, which is not match with the data in Table 1.
Response: This was corrected it was a typo error and now we have correct values for prestarter and starter diets – 0.22 and 0.2% respectively.
Line 106, what is the ‘J’ represent?
Response: It was a typo error, should be ‘I’ - Iodine
Line 242, it is inappropriate to pointed out that differences in response to growth might be linked to the source of phytase used. Many factors affect the growth of the piglets, and different protein source affect the activity of phytase. I suggest rewrite this sentence.
Response: We have corrected these sentences to avoid the statement that differences in growth performance were caused by a source of phytase. Naw we have the following sentences:
Differences in response to growth might be linked to the activity of phytase used, as observed by Guggenbuhl et al. [25], who showed that a greater inclusion of E. coli phytase was needed to obtain a significant increase in the daily gain of weaned piglets compared to Citrobacter braakii phytase or Buttauxiella phytase (diet containing 0.27% phytic-P and 0.55% total P).
Line 262-265, how did you come to this conclusion? Is there any reference?
Response: We have deleted these sentences.
There are many formatting errors in this article, such as the writing of "P-value"(P or P), the writing of vitamin K3(K3 or K3) on line 108 and so on. Please check them carefully. Dietary components in each group are not equal to 100% in Table 1. Please correct the data. The titles and notes of Tables 3, 4 and 7 are not consistent with other table formats. Please check these issues such as "centering, line height, and first line indentation". There are no Tables 5 and 6 in this article.
Tabel footnotes were corrected as well as table numeration and text centering etc. we corrected dietary components sum and new it is equal 100% or 99.99% (what is in line with results of formulation)
Some data do not meet the formula FCR = FI / BWG in Table 4. And Isn't day 0-42 of the NC + AppAT1 column in Table 4 29.47 ≈ 29.5? Why is 29.47≈29.4? Please correct the data in the table.
Response: Yes, but it is correct. E.g.: Originally BWG 0-21 was 12.358 (≈12.4), BWG 21-42 was 17.083 (≈17.1), 12.358 + 17.083 = 29.442 (≈29.4). FCR was calculated independently for each period and using individual data (FI and BWG) and that is the reason why we have some discrepancy between values.
It is not recommended to place Table 7 on two pages, please typeset reasonably. Reference to ‘X. Wu, Z. Ruan, Y. G. Zhang, Y. Q. Hou, Y. L. Yin*, T. J. Li, R. L. Huang, W. Y. Chu, X. F. Kong, B. Gao, L. X. Chen. True digestibility of P in different resources of P ingredients. Asian-Australasian Journal of Animal Science, 2008, 21(1):107-1
Response: Now the table is over one page and we used the above reference in the current version of the paper.
Reviewer 2 Report
Did you use the average TiO2 levels in diets to calculate results, or you used the TiO2 for each diet separately as presented in Table 3?
Line 266 - "The affinity of Ca and other multivalent cations ...". Is Ca multivalent cation?
Author Response
Dear Reviewer,
The authors would like to thank the reviewer for a helpful critique of the manuscript. The manuscript has been improved to reflect the required changes and these modifications are detailed below and directly in the text using track-changes mode.
Did you use the average TiO2 levels in diets to calculate results, or you used the TiO2 for each diet separately as presented in Table 3?
We used TiO2 for each diet separately
Line 266 - "The affinity of Ca and other multivalent cations ...". Is Ca multivalent cation?
This was delated and now we have the following sentence: The affinity of Ca and multivalent cations to greater phosphorylated inositol phosphates (InsP6) ensures the formation of phytate–mineral insoluble bonds [1].
Reviewer 3 Report
Experimental design is unclear; the piglet part of the methods refers to a complete block design (no blocking factors are mentioned), but the statistical analysis section is for a completely randomized design.
Results section unnecessarily repeats the data from the tables in the text. This section describes results as lower or higher irrespective of whether they are significantly different.
Conclusion as written does not accurately reflect the results since it does not specify to what the phytase diets are being compared. The data show that addition of phytases to a deficient diet restored parameters measured to those of the positive control. The exceptions are calcium ATTD and % of ash, which are higher in both phytase treatments than in both PC and NC.

Author Response
Dear Reviewer,
The authors would like to thank the reviewer for a helpful critique of the manuscript. The manuscript has been improved to reflect the required changes and these modifications are detailed below and directly in the text using track-changes mode.
Experimental design is unclear; the piglet part of the methods refers to a complete block design (no blocking factors are mentioned), but the statistical analysis section is for a completely randomized design.
Response: This was improved and now we have the following sentences:
The study was designed as a complete randomized with 12 replications of four diets: a positive control (PC) (T1), negative control (NC) (T2), and two experimental NC diets with the addition of phytases: appAT1 phytase (T3) and appAT2 phytase (T4), dosed at 250 FTU/kg.
The results section unnecessarily repeats the data from the tables in the text. This section describes results as lower or higher irrespective of whether they are significantly different.
Conclusion as written does not accurately reflect the results since it does not specify to what the phytase diets are being compared. The data show that the addition of phytases to a deficient diet restored parameters measured to those of the positive control. The exceptions are calcium ATTD and % of ash, which are higher in both phytase treatments than in both PC and NC.
Response: This has been corrected and now we have the following sentence in “introduction” section:
In the present study, two novel variants of phytases with highly improved thermal stability were obtained via protein engineering from the appA gene of E. coli [10]. Both strains differ in speed of phytate degradation and affinity for phytate (Vmax and KM respectively; in vitro Michaelis-Menten Kinetics).
Additionally, we have rewritten the conclusion section to reflect the above suggestion. Now we have:
The addition of two novel phytases, appAT1 and appAT2, to weaned piglets’ diets deficient in Ca and P at a dose of 250 FTU/kg/diet significantly improved the final BW, BWG, and FCR. The obtained data showed that the addition of phytases to a phosphorus-deficient diet restored performance parameters to those supplemented with the recommended phosphorus level. The ATTD of Ca and P was also significantly increased to values even higher to those observed in positive control pigs. However, no significant improvement in dry matter or crude protein digestion was noted.
Round 2
Reviewer 1 Report
None
Author Response
Dear Reviewer,
The authors would like to thank the reviewer for a helpful critique of the manuscript. The manuscript has been improved to reflect the required changes and these modifications are detailed below and directly in the text using track-changes mode.
We found two remarks in this review:
Moderate English changes required
The manuscript has undergone English language editing by MDPI, the certificate is available at the website.
Introduction section and conclusions Can be improved
Conclusion section was improved, we deleted some sentence and now we have: “The ATTD of Ca and P was also significantly increased to values observed in positive control pigs.”. The introduction section was improved during the last round and without specific comments, we don’t know what can be improved now.
Reviewer 3 Report
The authors continue to inappropriately describe results that are not significantly different as lower or higher. This not only statistically not justified, but also distracts from the results that are in fact significantly different.
Author Response
Dear Reviewer,
The authors would like to thank the reviewer for a helpful critique of the manuscript. The manuscript has been improved to reflect the required changes and these modifications are detailed below and directly in the text using track-changes mode.
We had only one comment - The authors continue to inappropriately describe results that are not significantly different as lower or higher. This not only statistically not justified, but also distracts from the results that are in fact significantly different.
We made fallowing changes (marked track changes mode in MS word):
L182: Piglets fed the NC diet had numerically the lowest final BW (36.9 kg) compared to the piglets receiving other diets.
L186-187: The end weight of the phytase-supplemented diets was greater compared to the PC diet, although this difference was not significant. – this sentence was deleted
L192-193: The FI was not significantly different between the different diets during the periods, but numerically the greatest FI was observed in the NC diet (Table 4).
L196-198: Pigs fed phytase-supplemented diets had a PC had similar FCR.
L205-207: The ATTD of the dry matter and crude protein was not affected by diet type (Table 5). The P ATTD was the lowest in piglets fed a NC diet (P<0.05 vs. other treatments; 27.6% vs 40.7%, 46.8%, and 48.4% for NC, PC, NC + appAT1, and NC + appAT2, respectively; Table 7).
L210-211: Similarly, there was a trend (P=0.069)toward a greater CP ATTD among the experimental diets.
L211-213: The lowest Ca ATTD was determined for piglets fed the unsupplemented NC diet (62.9%), but there were no differences with the PC (64.4%) diet (P>0.05).
L227-228: There were no significant differences between total ash weights across diets, although numerically the lowest ash weight was observed for the unsupplemented NC diet.